# DriveVLM: The Convergence of Autonomous Driving and Large Vision-Language Models

Xiaoyu Tian[1*]   Junru Gu[1*]   Bailin Li[2*]   Yicheng Liu[1*]   Yang Wang[2]   Zhiyong Zhao[2]

Kun Zhan[2]   Peng Jia[2]   Xianpeng Lang[2]   Hang Zhao[1†]

[1]IIIS, Tsinghua University   [2]Li Auto

**Abstract:** A primary hurdle of autonomous driving in urban environments is understanding complex and long-tail scenarios, such as challenging road conditions and delicate human behaviors. We introduce DriveVLM, an autonomous driving system leveraging Vision-Language Models (VLMs) for enhanced scene understanding and planning capabilities. DriveVLM integrates a unique combination of reasoning modules for scene description, scene analysis, and hierarchical planning. Furthermore, recognizing the limitations of VLMs in spatial reasoning and heavy computational requirements, we propose DriveVLM-Dual, a hybrid system that synergizes the strengths of DriveVLM with the traditional autonomous driving pipeline. Experiments on both the nuScenes dataset and our SUP-AD dataset demonstrate the efficacy of DriveVLM and DriveVLM-Dual in handling complex and unpredictable driving conditions. Finally, we deploy the DriveVLM-Dual on a production vehicle, verifying it is effective in real-world autonomous driving environments.

**Keywords:** Autonomous Driving, Vision Language Model, Dual System

## 1   Introduction

Autonomous driving, with its great promise to revolutionize transportation, has been an active research area over the past two decades. A primary hurdle to a fully autonomous driving system is scene understanding [1], which involves navigating complex, unpredictable scenarios such as adverse weather, intricate road layouts, and unforeseen human behaviors.

Existing autonomous driving systems, typically comprising 3D perception, motion prediction, and planning, struggle with these scene understanding challenges. Specifically, 3D perception [2, 3, 4, 5] is limited to detecting and tracking familiar objects, omitting rare objects and their unique attributes; motion prediction [6, 7, 8, 9, 10] and planning [11, 12, 13] focus on trajectory-level actions, often neglecting the decision-level interactions between objects and vehicles.

We introduce **DriveVLM**, a novel autonomous driving system that aims at these scene understanding challenges, capitalizing on the recent Vision-Language Models (VLMs) [14, 15, 16, 17] which have demonstrated exceptional prowess in visual comprehension and reasoning. Specifically, DriveVLM contains a Chain-of-Though (CoT) process with three key modules: *scene description*, *scene analysis*, and *hierarchical planning*. The scene description module linguistically depicts the driving environment and identifies critical objects in the scene; the scene analysis module delves into the characteristics of the critical objects and their influence on the ego vehicle; the hierarchical planning module formulates plans step-by-step, from meta-actions and decision descriptions to waypoints. These modules respectively correspond to the components of the traditional *perception-prediction-planning* pipeline, but the difference is that these modules tackle *object perception*, *intention-level prediction* and *task-level planning*, which were extremely challenging to cope with in the past.

---

*Equal contribution. Listing order is random.
†Corresponding to: hangzhao@mail.tsinghua.edu.cn
Project Page: https://tsinghua-mars-lab.github.io/DriveVLM/

8th Conference on Robot Learning (CoRL 2024), Munich, Germany.

While VLMs excel in visual understanding, they have limitations in spatial grounding and reasoning, and their computational intensity poses challenges for onboard inference speed. Therefore we further propose **DriveVLM-Dual**, a hybrid system that combines the strengths of both DriveVLM and traditional systems. DriveVLM-Dual optionally integrates DriveVLM with traditional 3D perception and planning modules, such as 3D object detectors, occupancy networks, and motion planners, enabling the system to achieve 3D grounding and high-frequency planning abilities. This dual system design, akin to the human brain's slow and fast thinking processes, adapts efficiently to varying complexity in driving scenarios.

Meanwhile, we formally define the scene understanding and planning (SUP) task, and propose new evaluation metrics to assess the scene analysis and meta-action planning capabilities of DriveVLM and DriveVLM-Dual. We carry out a comprehensive data mining and annotation pipeline to construct an in-house SUP-AD dataset for the SUP task. Extensive experiments on both the nuScenes dataset and our own dataset demonstrate the superior performance of DriveVLM, particularly in few-shot scenarios. Furthermore, DriveVLM-Dual exceeds state-of-the-art end-to-end motion planning methods. We have also deployed the model on a production vehicle, confirming that DriveVLM-Dual is effective in real-world autonomous driving environments. Additionally, we have included a demo in the supplementary materials.

In summary, the contribution of this paper is three-fold:

1. We introduce DriveVLM, a novel autonomous driving system that leverages VLMs for effective scene understanding and planning. We further introduce DriveVLM-Dual, a hybrid system that incorporates DriveVLM and a traditional autonomous pipeline, which achieves improved spatial reasoning and real-time planning capabilities.

2. We present a comprehensive data mining and annotation pipeline to construct a scene understanding and planning dataset (SUP-AD), together with metrics for evaluation.

3. We have successfully deployed DriveVLM-Dual system in a production vehicle and test various effective strategies for accelerating VLM deployment in real driving scenarios.

## 2 Related Works

**Vision-Language Models (VLMs).** Recently, there has been a surge in research on large Vision-Language Models (VLMs), exemplified by works such as MiniGPT-4 [16], LLaVA [17], Qwen-VL [18], and others [19, 14, 20, 21]. VLMs can be used in various scenarios, especially robotics [22, 23, 24, 25, 26], where VLMs output corresponding actions that can be high-level instructions [22] or low-level robot actions [24]. DriveVLM focuses on utilizing VLMs to assist in autonomous driving, thereby establishing a novel framework. A Concurrent work [15] shares a similar motivation.

**Learning-based Planning.** The integration of learning frameworks into motion planning has been an active area of research since Pomerleau [11] pioneering contributions. One promising line of work is Reinforcement learning and imitation learning [27, 28, 29]. These methods can learn an end-to-end planning policy that directly maps raw sensory inputs to control actions [29]. Several works [30, 31, 32, 33] improve interpretability by explicitly building dense cost maps derived from learning-based modules. A recent trend involves training multiple blocks in an end-to-end fashion [32, 33, 34, 35]. These methods enhance overall performance, but rely on backpropagation from future trajectory predictions loss in a less interpretable decision-making process [36].

**Driving Caption Datasets.** Recent works [37, 15, 38] argue that language captions are an important medium to connect human knowledge with the driving objective, helping to inform decisions and actions. Refer-KITTI [39] annotates objects in the KITTI dataset [40] with language prompts that can reference a collection of objects. Talk2Car [41], NuPrompt [42] and nuScenes-QA [43] introduce free-form captions and QA annotation to the nuScenes dataset [44]. BDD-X [45] and BDD-OIA [46] offer datasets with language explanations for the ego vehicle's actions or traffic scenarios [47] [48]. These datasets offer scenes for natural language use, but lack sufficient data on critical safety scenarios in self-driving systems.

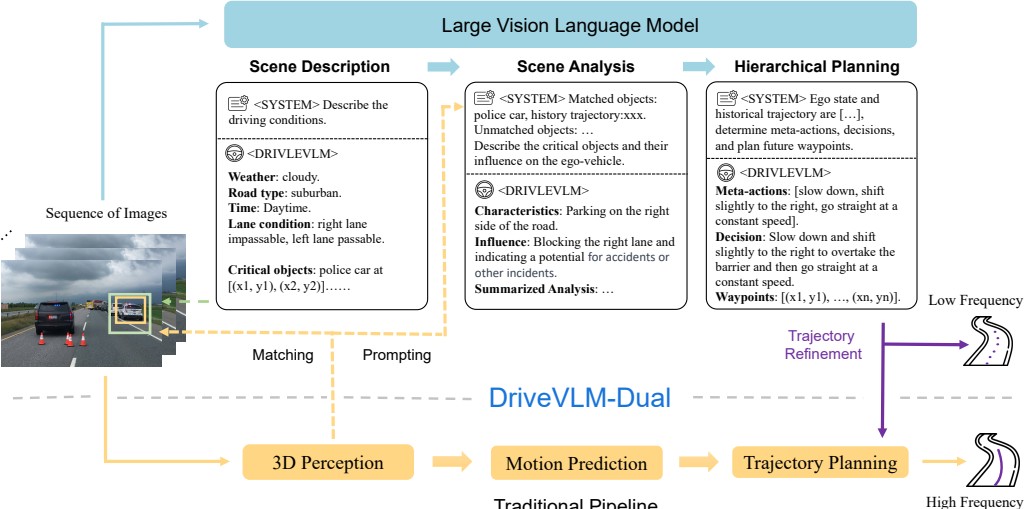

Figure 1: **DriveVLM and DriveVLM-Dual model pipelines.** DriveVLM takes images as input and, through a Chain-of-Thought (CoT) mechanism, outputs scene description, scene analysis, and hierarchical planning results. DriveVLM-Dual further incorporates traditional 3D perception and trajectory planning modules to achieve spatial reasoning capability and real-time trajectory planning.

# 3 DriveVLM

DriveVLM takes images as input and outputs planning results through a CoT reasoning process, as illustrated in Figure 1. The reasoning process can be divided into three modules: scene description (Section 3.1), scene analysis (Section 3.2), and hierarchical planning (Section 3.3). For real-world deployment, we propose a hybrid system, DriveVLM-Dual, in Section 3.4, which combines DriveVLM and the traditional pipeline, leveraging the strengths of both approaches.

## 3.1 Scene Description

The scene description module identifies driving environment description and critical objects.

**Environment Description.** Driving environments, such as weather and road conditions, have a non-negligible impact on driving difficulty. Therefore, the model is first prompted to output a linguistic description $E$ of the driving environment, including several conditions: $E = \{E_{\text{weather}}, E_{\text{time}}, E_{\text{road}}, E_{\text{lane}}\}$, each representing a crucial aspect of the driving environment. The weather component, $E_{\text{weather}}$, spans conditions from sunny to snowy, affecting visibility and traction. The time component, $E_{\text{time}}$, distinguishes between daytime and nighttime, impacting driving strategies due to visibility changes. Road types, $E_{\text{road}}$, such as urban or highway, introduce different challenges, while lane conditions, $E_{\text{lane}}$, focus on current lane positioning and possible maneuvers, crucial for safe driving decisions.

**Critical Object Identification.** Unlike traditional autonomous driving perception modules, which detect all objects within a specific range, we solely focus on identifying *critical objects* that are most likely to influence the current scenario, inspired by human cognitive processes during driving. Each *critical object*, denoted as $O_c$, contains two attributes: the object category $c$ and its approximate bounding box coordinates $b(x1, y1, x2, y2)$ on the image. Taking advantage of the pre-trained vision encoder, DriveVLM can identify long-tail *critical objects* that may elude typical 3D object detectors, such as road debris or unusual animals.

## 3.2 Scene Analysis

In the traditional autonomous driving pipeline, the prediction module typically concentrates on forecasting the future trajectories of objects. The emergence of advanced vision-language models has provided us with the ability to perform a more comprehensive analysis of the current scene. The scene-level analysis summarizes all the critical objects together with the environmental description.

This summary gives a comprehensive understanding of the scene, and is fed into the following planning module.

**Critical Object Analysis.** DriveVLM characterizes critical objects in three aspects: **static attributes** $C_s$, **motion states** $C_m$, and **particular behaviors** $C_b$. Static attributes $C_s$ describe inherent properties of objects, such as a roadside billboard's visual cues or a truck's oversized cargo, which are critical in preempting and navigating potential hazards. Motion states $C_m$ describe an object's dynamics over a period, including position, direction, and action—characteristics that are vital in predicting the object's future trajectory and potential interactions with the ego vehicle. Particular behaviors $C_b$ refer to special actions or gestures of an object that could directly influence the ego vehicle's next driving decisions. We do not require the model to analyze all three characteristics for all objects. In practice, only one or two characteristics apply to a critical object. Upon analyzing these characteristics, DriveVLM then predicts the potential influence $I$ of each critical object on the ego vehicle.

## 3.3 Hierarchical Planning

The scene-level summary is then combined with the route, ego pose and velocity to form a prompt for planning. Finally, DriveVLM progressively generates driving plans, in three stages: meta-actions, decision description, and trajectory waypoints.

**Meta-actions $A$.** A meta-action, denoted as $a_i$, represents a short-term decision of the driving strategy. These actions fall into 17 categories, including but not limited to acceleration, deceleration, turning left, changing lanes, minor positional adjustments, and waiting. To plan the ego vehicle's future maneuver over a certain period, we generate a sequence of meta-actions.

**Decision Description $D$.** Decision description $D$ articulates the more fine-grained driving strategy the ego vehicle should adopt. It contains three elements: Action $\mathcal{A}$, Subject $\mathcal{S}$, and Duration $\mathcal{D}$. *Action* pertains to meta actions such as 'turn', 'wait', or 'accelerate'. *Subject* refers to the interacting object, such as a pedestrian, a traffic signal, or a specific lane. *Duration* indicates the temporal aspect of the action, specifying how long it should be carried out or when it should start.

**Trajectory Waypoints $W$.** Upon establishing the decision description $D$, our next phase involves the generation of corresponding trajectory waypoints. These waypoints, denoted by $W = \{w_1, w_2, ..., w_n\}$, $w_i = (x_i, y_i)$, depict the vehicle's path over a certain future period with predetermined intervals $\Delta t$. We map these numerical waypoints into language tokens for auto-regressive generation.

## 3.4 DriveVLM-Dual

To mitigate the challenges of high latency and imprecise spatial and motion understanding in VLMs, we propose DriveVLM-Dual, a collaboration between DriveVLM and the traditional autonomous driving system. This novel approach involves two key strategies: incorporating 3D perception for critical object analysis, and high-frequency trajectory refinement.

**Integrating 3D Perception.** We represent objects detected by a 3D detector as $O_{3D} = \{c_{3D}^i, b_{3D}^i\}$, where $b_{3D}^i$ denotes the $i$-th bounding box and $c_{3D}^i$ denotes its category. These 3D bounding boxes are then back-projected onto 2D images to derive corresponding 2D bounding boxes $b_{2D}^i$. We conduct IoU matching between these 2D bounding boxes $b_{2D}^i$ and $b_c^j$. $b_c^j$ are the bounding boxes of previously identified critical objects $O_{\text{critical}} = \{c_c^j, b_c^j\}$. We classify critical objects that meet a certain approximate IoU threshold and belong to the same category as matched critical objects $O_c^{\text{matched}}$, defined as

$$O_c^{\text{matched}} = \{c_c^j, b_c^j\}, \quad \text{if } c_c^j = c_{2D}^i \text{ and aIoU}(b_c^j, b_{2D}^i) > \tau, \text{ where aIoU}(b_c^j, b_{2D}^i) = \frac{S_{b_c^j \cap b_{2D}^i}}{S_{b_{2D}^i}},$$

Those critical objects without a corresponding match in the 3D data are noted as $O_c^{\text{unmatched}}$.

In the scene analysis module, for $O_c^{\text{matched}}$, the center coordinates, orientations, and historical trajectories of the corresponding 3D objects are used as language prompts for the model, assisting in

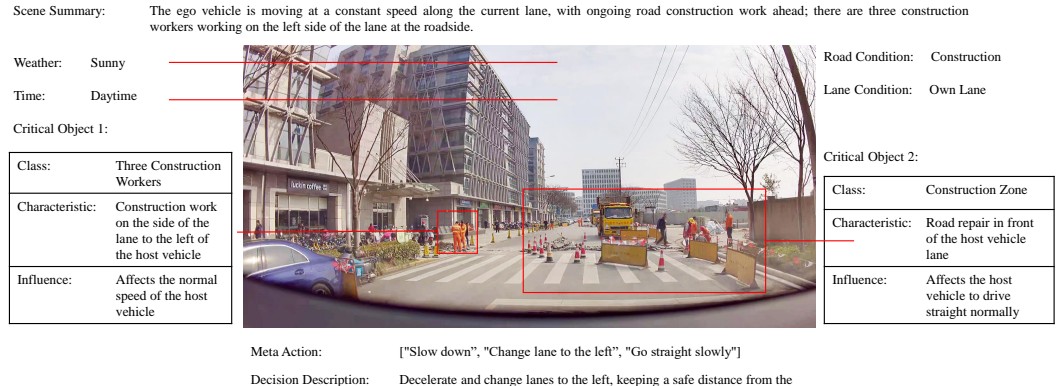

| Scene Summary: | The ego vehicle is moving at a constant speed along the current lane, with ongoing road construction work ahead; there are three construction workers working on the left side of the lane at the roadside. |
| --- | --- |

| Weather: | Sunny | | Road Condition: | Construction |
| --- | --- | --- | --- | --- |
| Time: | Daytime | | Lane Condition: | Own Lane |

Critical Object 1:

| Class: | Three Construction Workers |
| --- | --- |
| Characteristic: | Construction work on the side of the lane to the left of the host vehicle |
| Influence: | Affects the normal speed of the host vehicle |

Critical Object 2:

| Class: | Construction Zone |
| --- | --- |
| Characteristic: | Road repair in front of the host vehicle lane |
| Influence: | Affects the host vehicle to drive straight normally |

| Meta Action: | ["Slow down", "Change lane to the left", "Go straight slowly"] |
| --- | --- |
| Decision Description: | Decelerate and change lanes to the left, keeping a safe distance from the construction workers on the left front side. |

Figure 2: **An annotated sample of the SUP-AD dataset.**

object analysis. Conversely, for $O_c^{\text{unmatched}}$, analysis relies solely on the language tokens derived from the image. This design enables DriveVLM-Dual to understand the locations and motions of critical objects more accurately, enhancing the overall performance.

**High-frequency Trajectory Refinement.** To achieve real-time, high-frequency inference capabilities, we integrate it with a conventional planner to form a slow-fast dual system, combining the advanced capabilities of DriveVLM with the efficiency of traditional planning methods. After obtaining a trajectory from DriveVLM at low frequency, denoted as $W_{\text{slow}}$, we take it as a reference trajectory for a classical planner for high-frequency trajectory refinement. In the case of an optimization-based planner, $W_{\text{slow}}$ serves as the initial solution for the optimization solver. For a neural network-based planner, $W_{\text{slow}}$ is used as an input query, combined with additional input features $f$, and then decoded into a new planning trajectory denoted as $W_{\text{fast}}$. The formulation of this process can be described as:

$$W_{\text{fast}} = \text{Planner}([W_{\text{slow}}, f]). \tag{1}$$

This refinement step ensures that the trajectory produced by DriveVLM-Dual (1) achieves higher trajectory quality, and (2) meets real-time requirements. In practice, the two branches operate asynchronously in a slow-fast manner, where the planner module in the traditional autonomous driving branch can selectively receive trajectory from the VLM branch as additional input.

## 4 Task and Dataset

To fully exploit the potential of DriveVLM and DriveVLM-Dual in handling complex and long-tail driving scenarios, we formally define a task called *Scene Understanding for Planning* (Section 4.1), together with a set of evaluation metrics (Section 4.2). Furthermore, we propose a data mining and annotation protocol to curate a scene understanding and planning dataset (Section 4.3).

### 4.1 Task Definition

The input of Scene Understanding for Planning task comprises multi-view videos $\mathcal{V}$ from surrounding cameras and optionally 3D perception results $\mathcal{P}$ from a perception module. And its output includes the following components: **Scene Description** $s$: Composed of different conditions, object-level analysis and scene-level summary; **Meta Actions** $A$: A sequence of actions representing task-level maneuvers; **Decision Description** $D$: A detailed account of the driving decisions; **Trajectory Waypoints** $W$: The waypoints outlining the planned trajectory of the ego vehicle.

### 4.2 Evaluation Metrics

To comprehensively evaluate a model's performance, we care about its interpretation of the driving scene and the decisions made. Therefore, our evaluation has two aspects: scene description/analysis evaluation and meta-action evaluation. In the **Scene description/analysis evaluation**, we leverage

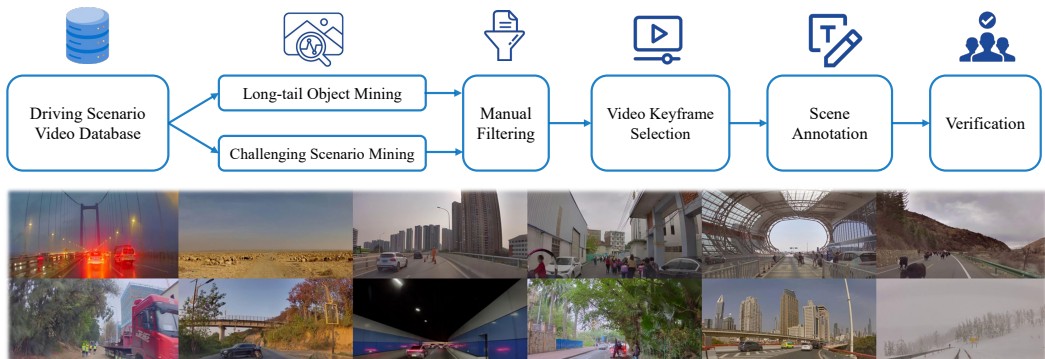

Figure 3: **The proposed data mining and annotation pipeline for constructing the scene understanding and planning dataset.** Scenario examples randomly sampled from the dataset (below) demonstrate the diversity and complexity of the dataset.

a structured evaluation approach by prompting the GPT-4 to extract individual pieces of information from each scene description. Subsequently, we ask the GPT-4 to score and aggregate the results based on the matching status of each extracted piece of information. For **Meta-action evaluation**, we focus on the sequence of decision-making actions defined as meta-actions, comparing these sequences generated by the model against a ground truth using dynamic programming. To ensure the robustness of our evaluation, we use GPT-4 to generate semantically equivalent alternatives to the ground truth, with the highest similarity score determining the final decision-making accuracy. Further details on these metrics are elaborated in Appendix B.

### 4.3  Dataset Construction

We propose a comprehensive data mining and annotation pipeline, shown in Figure 3, to construct a *Scene Understanding for Planning (SUP-AD) Dataset* for the proposed task. This process begins with **long-tail object mining** and **challenging scenario mining** from a large database to collect special objects and scenes like weird-shaped vehicles and road debris. For each scene, **Keyframe selection** is performed to identify crucial moments for decision-making, which are then annotated using a specialized **Scene Annotation** tool. Details of the construction process and dataset statistics are available in the Appendix A.

## 5  Experiments

### 5.1  Settings

We test DriveVLM and DriveVLM-Dual on our proposed SUP-AD dataset and nuScenes dataset [44]. The setup details of our model and training details are listed in Appendix C.

**SUP-AD Dataset.** The SUP-AD dataset is a dataset built by our proposed data mining and annotation pipeline. It is divided into train, validation, and test splits with a ratio of 7.5 : 1 : 1.5. We train models on the training split and use our proposed scene description and meta-action metrics to evaluate model performance on the test split. We also employ co-tuning with additional datasets to ensure the generalization of the LLM is not compromised. For details, see Appendix C.

**nuScenes Dataset.** The nuScenes dataset is a large-scale driving dataset of urban scenarios with 1000 scenes, where each scene lasts about 20 seconds. Following previous works [34, 49], we adopt Displacement Error (DE) and Collision Rate (CR) as metrics to evaluate models' performance.

### 5.2  Main Results

**SUP-AD.** We present the performance of our proposed DriveVLM with several large vision-language models and compare them with GPT-4V, as shown in Table 1. DriveVLM, utilizing Qwen-VL as its backbone, achieves the best performance due to its strong capabilities in question answer-

Table 1: **Results on the test set of our proposed SUP-AD dataset.** [†]: Using the official API of GPT-4V. For Lynx and CogVLM, we utilize the training split for fine-tuning purposes. In contrast, for GPT-4V, we employ in-context learning.

| Method | Scene Description | Meta-actions |
|---|---|---|
| Fine-tuning w/ Lynx [14] | 0.46 | 0.15 |
| Fine-tuning w/ CogVLM [21] | 0.49 | 0.22 |
| GPT-4V[†] [50] | 0.38 | 0.19 |
| DriveVLM w/ Qwen | **0.71** | **0.37** |

Table 2: **Planning results on the nuScenes validation dataset.** DriveVLM-Dual achieves the best performance. [†] denotes cooperating with VAD [49].

| Method | L2 (m) ↓ | | | | Collision (%) ↓ | | | |
|---|---|---|---|---|---|---|---|---|
| | 1s | 2s | 3s | Avg. | 1s | 2s | 3s | Avg. |
| NMP [30] | - | - | 2.31 | - | - | - | 1.92 | - |
| SA-NMP [30] | - | - | 2.05 | - | - | - | 1.59 | - |
| FF [32] | 0.55 | 1.20 | 2.54 | 1.43 | 0.06 | 0.17 | 1.07 | 0.43 |
| EO [51] | 0.67 | 1.36 | 2.78 | 1.60 | 0.04 | 0.09 | 0.88 | 0.33 |
| ST-P3 [52] | 1.33 | 2.11 | 2.90 | 2.11 | 0.23 | 0.62 | 1.27 | 0.71 |
| UniAD [34] | 0.48 | 0.96 | 1.65 | 1.03 | 0.05 | 0.17 | 0.71 | 0.31 |
| VAD-Base [49] | 0.17 | 0.34 | 0.60 | 0.37 | 0.07 | 0.10 | 0.24 | 0.14 |
| DriveVLM | 0.18 | 0.34 | 0.68 | 0.40 | 0.10 | 0.22 | 0.45 | 0.27 |
| **DriveVLM-Dual**[†] | **0.15** | **0.29** | **0.48** | **0.31** | **0.05** | **0.08** | **0.17** | **0.10** |

ing and flexible interaction compared to the other open-source VLMs. Although GPT-4V exhibits robust capabilities in vision and language processing, its inability to undergo fine-tuning, restricting it solely to in-context learning, often results in the generation of extraneous information during scene description tasks. Under our evaluation metric, the additional information is frequently classified as hallucination, consequently leading to lower scores.

**nuScenes.** As shown in Table 2, DriveVLM-Dual achieves state-of-the-art performance on the nuScenes planning task when cooperating with VAD. It demonstrates that our method, although tailored for understanding complex scenes, also excels in ordinary scenarios.

## 5.3 Ablation Study

**Model Design.** To better understand the significance of our designed modules in DriveVLM, we conduct ablations on different combinations of modules, as shown in Table 3. The inclusion of critical object analysis enables our model to identify and prioritize important elements in the driving environment, enhancing the decision-making accuracy for safer navigation. Integrating 3D perception data, our model gains a refined understanding of the surroundings and achieves precise predictions.

**Traditional AD Pipeline.** To demonstrate the generalization of our dual system design, we test DriveVLM-Dual with different traditional autonomous driving pipelines on the validation set of nuScenes. As illustrated in Table 4, our proposed DriveVLM-Dual adapts well to different traditional AD pipelines. While a standalone MLP method shows a notable performance gap compared to VAD, both variants of DriveVLM-Dual achieve nearly identical performance, underscoring the efficacy and robustness of our dual system design.

## 5.4 Qualitative Results

Qualitative results of DriveVLM are shown in Figure 4. In Figure 4a, DriveVLM accurately predicts the current scene conditions and incorporates well-considered planning decisions regarding the cyclist approaching us. In Figure 4b, DriveVLM effectively comprehends the gesture of the traffic police ahead, signaling the ego vehicle to proceed, and also considers the person riding a tricycle on the right side, thereby making sensible driving decisions. These qualitative results demonstrate our model's exceptional ability to understand complex scenarios and make suitable driving plans. More visualization of our model's output is shown in the Appendix E.

Table 3: **Ablations of design choices on the validation set of nuScenes.** "Base" refers to only indicating the hierarchical planning results without our proposed CoT inference. "CO" represents the addition of critical object analysis. "3D" denotes the inclusion of 3D perception results as an auxiliary language prompt.

| ID | Base | CO | 3D | L2 (m) ↓ | | | | Collision (%) ↓ | | | |
|---|---|---|---|---|---|---|---|---|---|---|---|
| | | | | 1s | 2s | 3s | Avg. | 1s | 2s | 3s | Avg. |
| 1 | ✓ | | | 0.19 | 0.41 | 0.89 | 0.49 | 0.16 | 0.28 | 0.63 | 0.36 |
| 2 | ✓ | ✓ | | 0.20 | 0.38 | 0.75 | 0.44 | 0.15 | 0.29 | 0.61 | 0.35 |
| 3 | ✓ | ✓ | ✓ | 0.18 | 0.34 | 0.68 | 0.40 | 0.10 | 0.22 | 0.45 | 0.27 |

Table 4: **Ablations of traditional autonomous driving pipeline in DriveVLM-Dual.** MLP stands for methods similar to AD-MLP [53].

| Method | L2 (m) ↓ | | | | Collision (%) ↓ | | | |
|---|---|---|---|---|---|---|---|---|
| | 1s | 2s | 3s | Avg. | 1s | 2s | 3s | Avg. |
| UniAD [34] | 0.48 | 0.96 | 1.65 | 1.03 | 0.05 | 0.17 | 0.71 | 0.31 |
| w/ DriveVLM-Dual | 0.17 | 0.37 | 0.63 | 0.39 | 0.08 | 0.18 | 0.35 | 0.20 |
| MLP | 0.25 | 0.46 | 0.62 | 0.44 | 0.14 | 0.18 | 0.28 | 0.20 |
| w/ DriveVLM-Dual | 0.14 | 0.35 | 0.30 | 0.31 | 0.09 | 0.13 | 0.18 | 0.13 |
| VAD [49] | 0.17 | 0.34 | 0.60 | 0.37 | 0.07 | 0.10 | 0.24 | 0.14 |
| w/ DriveVLM-Dual | 0.15 | 0.29 | 0.48 | 0.31 | 0.05 | 0.08 | 0.17 | 0.10 |

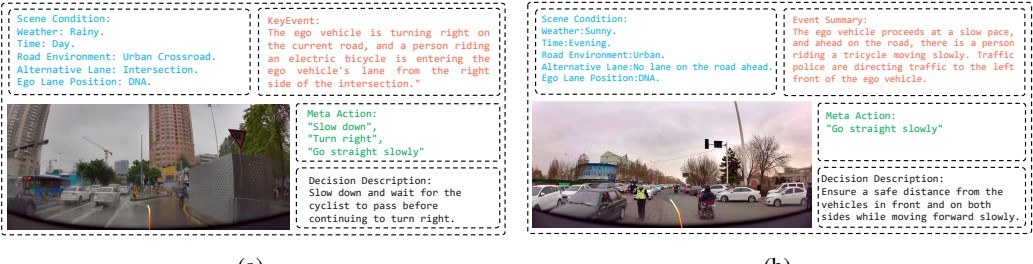

Figure 4: **Qualitative results of DriveVLM.** The orange curves represent the model's planned future trajectories for the next 3 seconds.

## 6 Onboard Deployment and Testing

We deploy DriveVLM-Dual on an autonomous vehicle equipped with two OrinX processors, with the fast end-to-end driving system on OrinX-1 and the slower DriveVLM on OrinX-2. The total computational power is 508 TOPS (including both sparse and DLA computational power). Given the vehicle's hardware constraints, we select models with fewer than 4 billion parameters for real-time inference. The Qwen series [18] demonstrate superior inference speed due to its wide and shallow architecture, meeting the vehicle's speed requirements with an average total time for prefill and decode within 400ms. For the visual encoder, we choose the SigLIP-L-384 [54] model with PE interpolation fine-tuning, optimizing for high resolution and fine-grained understanding. To handle the computational load from high-resolution images, we use LDPNetv2 [55], reducing image tokens by 75% without compromising performance, and cropped non-informative regions from images. To better assess object motion, we employ a memory bank strategy [56] for storing visual features from historical frames, enhancing temporal context with SE [57] blocks for weighted fusion. We utilize the Eagle [58] speculative sampling technique, achieving a 2.7x speedup in decode latency with our custom inference framework on the Orin chip, ensuring feasible real-time deployment for autonomous driving. Through these designs and optimizations, the DriveVLM can achieve a real-time inference speed of approximately 400ms per inference. The end-to-end fast system has a real-time speed of 130 ms per inference, and since the two systems run asynchronously, the overall system latency is about 130 ms. For detailed deployment techniques and model performance, please refer to Appendix D.

## 7 Limitations

Our VLM model is obtained by continuing pretraining and fine-tuning an existing pretrained VLM. Consequently, the upper bound of VLM's capabilities is inherently limited by that of the original pretrained VLM. Moreover, current VLMs exhibit certain deficiencies in spatial understanding, such as position and distance, which is why we integrate 3D perception results into VLM's input. Additionally, current VLMs are susceptible to "hallucination" issues, which can potentially lead to safety concerns. Completely eliminating such hallucinations remains a challenge, though this is an inherent limitation of VLMs and not specific to our method.

## 8 Conclusion

In summary, we introduce DriveVLM and DriveVLM-Dual. DriveVLM leverages VLMs, significantly progressing in interpreting complex driving environments. DriveVLM-Dual further enhances these capabilities by synergizing existing 3D perception and planning approaches, effectively addressing the spatial reasoning and computational challenges inherent in VLMs. Moreover, we define a scene understanding for planning task for autonomous driving, together with evaluation metrics and dataset construction protocol. DriveVLM and DriveVLM-Dual have surpassed the state-of-the-art methods on the public and our benchmarks, especially in handling intricate and dynamic scenarios. Finally, we have verified the effectiveness of DriveVLM-Dual through onboard deployment and testing on a production vehicle.

**Acknowledgments**

We thank Chenxu Hu at Tsinghua University for the help on paper writing. We thank Xu Bian, Hongkun Chen, Sui Cong, Chengze Guan, Mingyu Guo, Yue Jiang, Qi Jiang, Pengfei Ji, Wei Xiao, Dafeng Wei, Zijian Wang, Zhao Yang, Chenglong Zhao, Simeng Zhao, and Jian Zhou at Li Auto for their efforts in the experiments related to onboard deployment.

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

# A    SUP-AD Dataset

## A.1    Meta-actions

**Meta-action statistics.**    We use the meta-action sequence to formally represent the driving strategy. Meta actions are classified into 17 categories. We show the distribution of each meta-action being the first/second/third place in the meta-action sequence, as shown in Figure 5. It indicates that the meta-actions are quite diverse in the SUP-AD dataset. We also show the distribution of the length of meta-actions per scene in Figure 6. Most scenes contain two or three meta-actions, and a few scenes with complex driving strategies contain four or more meta-actions.

**Annotation of meta-actions.**    The meta-action sequence for each driving scene is manually annotated based on the actual driving strategy in the future frames. These meta-actions are designed to encompass a complete driving strategy and are structured to be consistent with the future trajectory of the ego vehicle. They can be divided into three primary classes:

1. **Speed-control actions.** Discerned from acceleration and braking signals within the ego state data, these actions include These actions can be discerned from acceleration and braking signals within the ego state data. They include *speed up*, *slow down*, *slow down rapidly*, *go straight slowly*, *go straight at a constant speed*, *stop*, *wait*, and *reverse*.

2. **Turning actions.** Deduced from steering wheel signals, these actions consist of *turn left*, *turn right*, and *turn around*.

3. **Lane-control actions.** Encompassing lane selection decisions, these actions are derived from a combination of steering wheel signals and either map or perception data. They involve *change lane to the left*, *change lane to the right*, *shift slightly to the left*, and *shift slightly to the right*.

## A.2    Scenario Categories

The SUP-AD dataset is comprised of 1,000 video clips of driving scenarios. As illustrated in Figure 7, it encompasses a wide range of driving scenarios, spanning over 40 categories. Below are explanations for some of the scenarios:

**AEB Data**: Automatic Emergency Braking (AEB) data.

**Road Construction**: A temporary work zone with caution signs, barriers, and construction equipment ahead.

**Close-range Cut-ins**: A sudden intrusion into the lane of the ego vehicle by another vehicle.

**Roundabout**: A type of traffic intersection where vehicles travel in a continuous loop.

**Animals Crossing Road**: Animals crossing the road in front of the ego vehicle.

**Braking**: Brake is pressed by human driver of the ego vehicle.

**Traffic Police Officers**: Traffic police officers managing and guiding traffic.

**Blocking Traffic Lights**: A massive vehicle obscuring the visibility of the traffic signal.

**Cutting into Other Vehicle**: Intruding into the lane of another vehicle ahead.

**Ramp**: A curved roadway that connects the main road to the branch road in highway.

**Debris on the Road**: Road with different kinds of debris.

**Narrow Roads**: Narrow roads that require cautious navigation.

**Pedestrians Popping Out**: Pedestrians popping out in front of the ego vehicle, requiring slowing down or braking.

**People on Bus Posters**: Buses with posters, which may interfere the perception system.

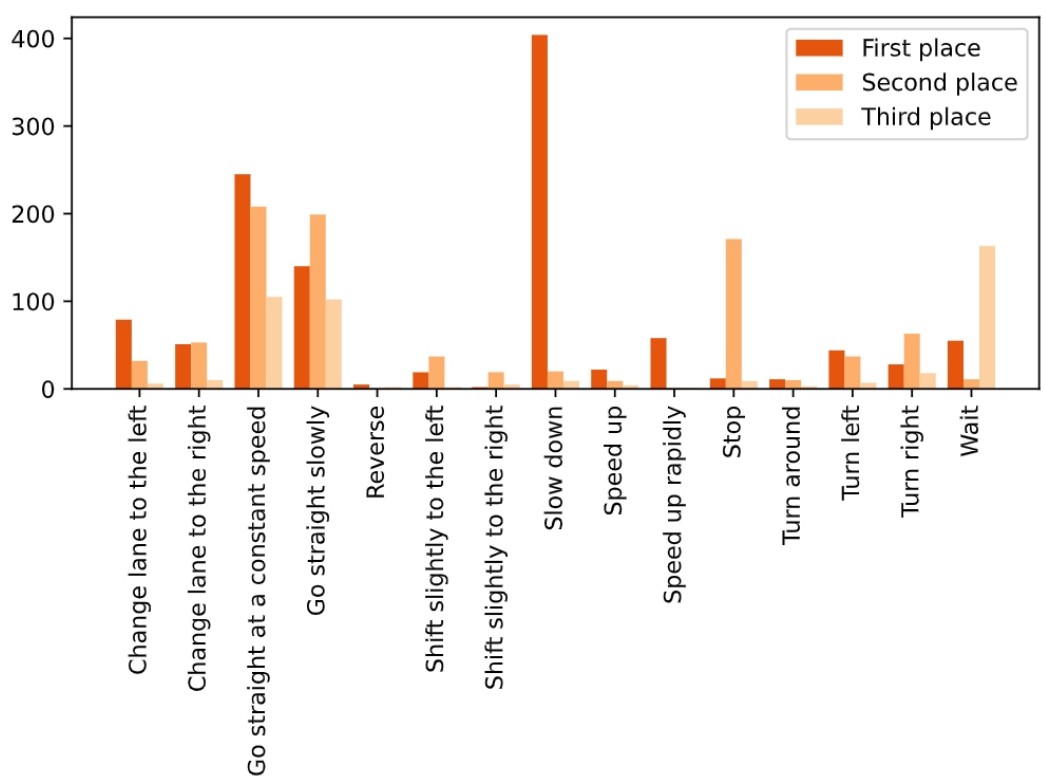

Figure 5: **Distribution of each meta action being the first, second, and third place of the meta action sequence, respectively.**

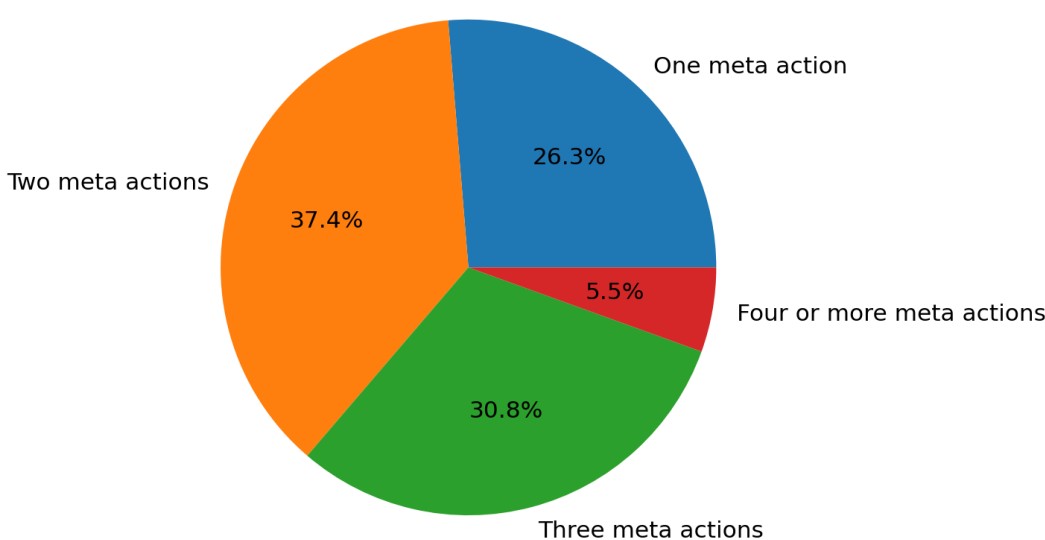

Figure 6: **Distribution of the length of meta actions per scene.**

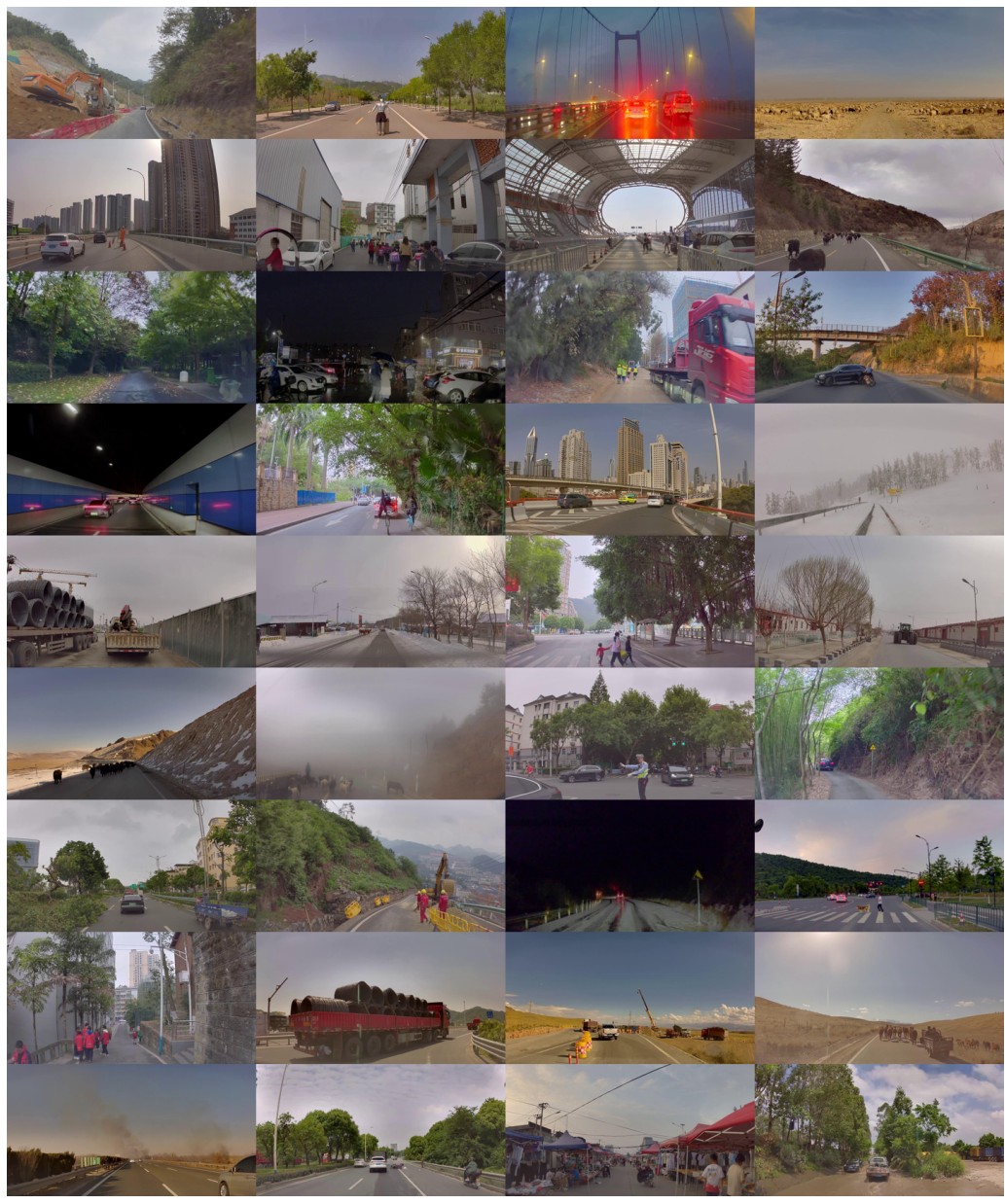

Figure 7: **Diverse driving scenarios in the SUP-AD dataset.**

**Merging into High Speed**: Driving from a low-speed road into a high-speed road, requiring speeding up.

**Barrier Gate**: Barrier gate that can be raised obstructing the road.

**Fallen Trees**: Fallen trees on the road, requiring cautious navigation to avoid potential hazards.

**Complex Environments**: Complex driving environments that requiring cautious navigation.

**Mixed Traffic**: A congested scenario where cars, pedestrians, and bicycles appear on the same or adjacent roadway.

**Crossing Rivers**: Crossing rivers by driving on the bridge.

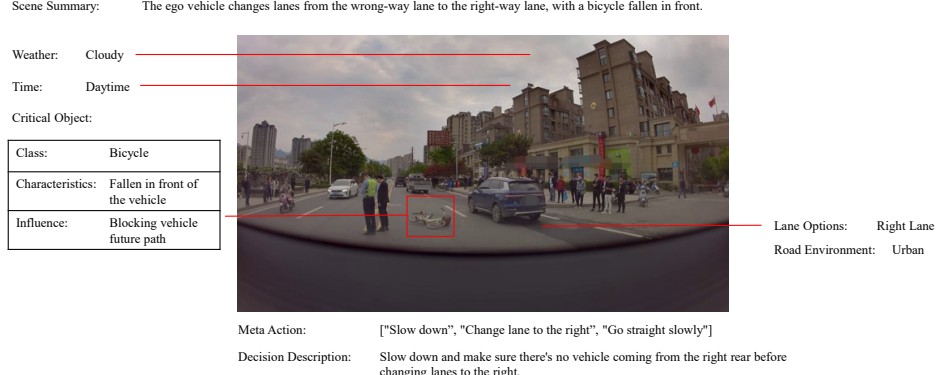

Figure 8: **An example of overturned bicycles and motorcycles in the SUP-AD dataset.** A bicycle has fallen in front of the ego vehicle, requiring the ego vehicle to change lanes.

**Screen**: Roads with screens on one side, which may interfere the perception system.

**Herds of Cattle and Sheep**: A rural road with herds of cattle and sheep, requiring careful driving to avoid causing distress to these animals.

**Vulnerable Road Users**: Road users which are more susceptible to injuries while using roads, such as pedestrians, cyclists, and motorcyclists.

**Road with Gallet**: A dusty road with gallet scattered across the surface.

The remaining scenario categories are: Motorcycles and Trikes, Intersection, People carrying Umbrella, Vehicles Carrying Cars, Vehicles Carrying Branches, Vehicles with Pipes, Strollers, Children, Tunnel, Down Ramp, Sidewalk Stalls, Rainy Day, Crossing Train Tracks, Unprotected U-turns, Snowfall, Large Vehicles Invading, Falling Leaves, Fireworks, Water Sprinklers, Potholes, Overturned Motorcycles, Self-ignition and Fire, Kites, Agricultural Machinery.

### A.3 Annotation Examples

We provide more examples of annotation contents in Figure 8, 9, 10, 11, 12, and 13. The scenario categories of these examples are overturned bicycles and motorcycles, herds of cattle and sheep, collapsed trees, crossing rivers, barrier gate, and snowfall respectively.

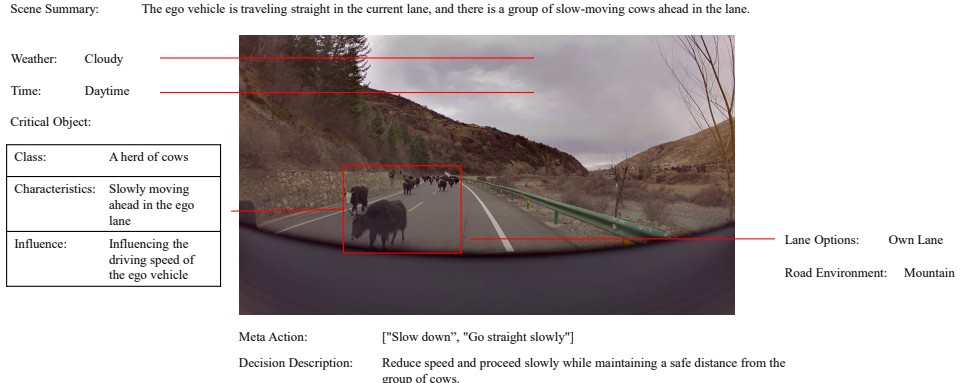

Figure 9: **An example of herds of cattle and sheep in the SUP-AD dataset.** A group of cattle move slowly in front of the ego vehicle, requiring the ego vehicle to proceed slowly and maintain a safe distance from the cattle.

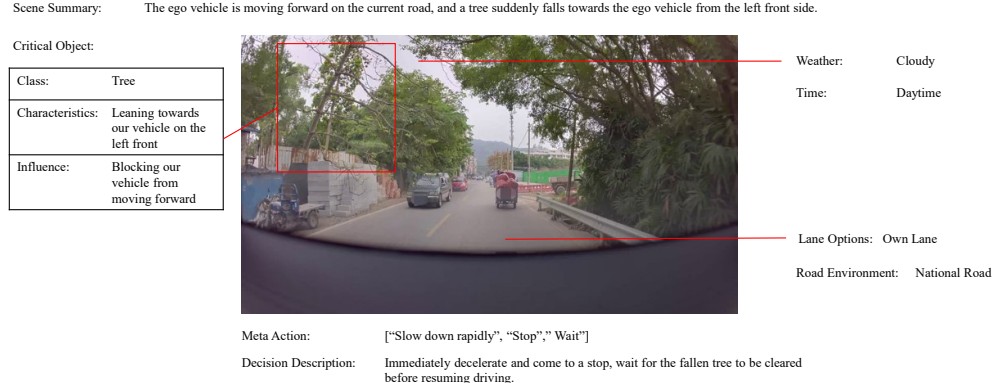

Figure 10: An example of collapsed trees in the SUP-AD dataset. A tree suddenly falls towards the ego vehicle, requiring the ego vehicle to decelerate immediately.

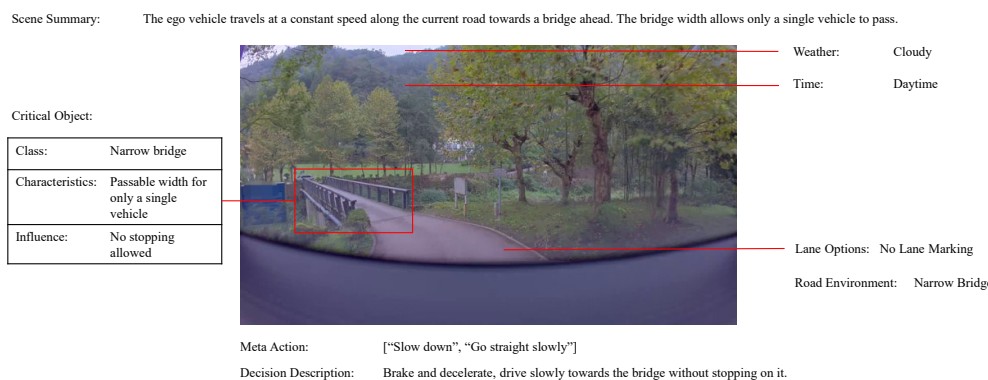

Figure 11: **An example of crossing rivers in the SUP-AD dataset.** The ego vehicle is going across a bridge of which width allows only a single vehicle to pass, requiring the ego vehicle to drive without stopping.

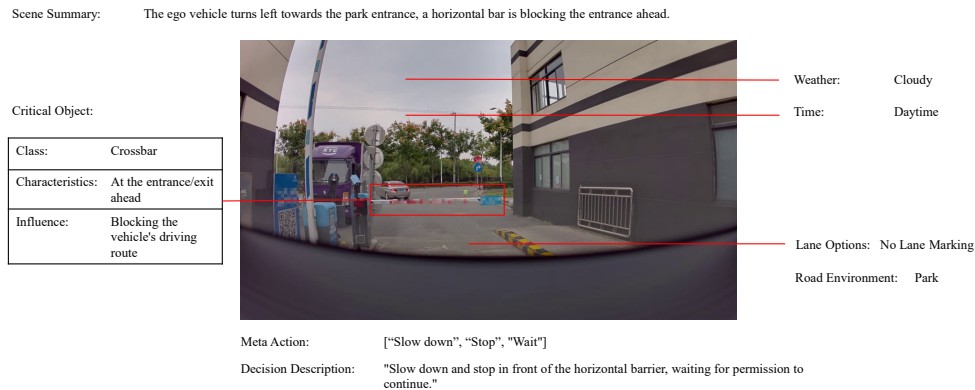

Figure 12: **An example of barrier gate in the SUP-AD dataset.** A horizontal barrier blocks the entrance of a park, requiring the ego vehicle to stop and wait for permission to continue.

Scene Summary: The ego vehicle is currently driving slowly along a snow-covered road, following the snow-free tire tracks on the road surface.

Critical Object: None

Weather: Snowy

Time: Daytime

Lane Options: No Lane Marking

Road Environment: Snow-covered Road

Meta Action: ["Go straight slowly"]

Decision Description: "Continue to move forward cautiously at a slow speed."

Figure 13: **An example of snowfall in the SUP-AD dataset.** Most of the road is covered by snow, requiring the ego vehicle to move forward cautiously by following the snow-free tire tracks.

# B  Evaluation Method

The ability of an autonomous driving system to accurately interpret driving scenes and make logical, suitable decisions is of paramount importance. As presented in this paper, the evaluation of VLMs in autonomous driving concentrates on two primary components: the evaluation of scene description/analysis and the evaluation of meta-actions.

## B.1  Scene Description/Analysis Evaluation

In terms of scene description/analysis evaluation, the process of interpreting and articulating driving scenes is subject to inherent subjectivity, as there are numerous valid ways to express similar descriptions textually, which makes it difficult to effectively evaluate the scene description using a fixed metric. To overcome this challenge, we utilize GPT-4 to evaluate the similarity between the scene descriptions generated by the model and the manually annotated ground truth. Initially, we prompt GPT-4 to extract individual pieces of information from each scene description. Subsequently, we score and aggregate the results based on the matching status of each extracted piece of information.

The ground truth labels for scene descriptions encompass both environment descriptions and event summaries. Environmental condition description includes weather conditions, time conditions, road environment, and lane conditions. Event summaries are the characteristics and influence of critical objects. We employ GPT-4 to extract unique key information from both environment descriptions and event summaries. The extracted information is then compared and quantified. Each matched pair is assigned a score, which is estimated based on the extent of the matching, whether complete, partial, or absent. Instances of hallucinated information incur a penalty, detracting from the overall score. The aggregate of these scores constitutes the scene description score.

$$\text{Score} = \frac{1.0 \times n_{\text{matched}} + 0.5 \times n_{\text{partial}}}{n_{\text{gt}}} \tag{2}$$
$$- \frac{0.25 \times n_{\text{hallucination}}}{n_{\text{gt}}}$$

The prompt for GPT-4 in evaluating scene descriptions is carefully designed, as shown in Table 5. Initially, a role prompt is employed to establish as an intelligent and logical evaluator, possessing a comprehensive understanding of appropriate driving styles. This is followed by specifying the input format, which informs GPT-4 that its task involves comparing an output description with a ground truth description. This comparison is based on the extraction and analysis of key information from

**System Prompt:**
You are a smart and logical evaluator with extensive driving experience, and you will try your best to give reasonable and logical evaluation result.

**Input Prompt:**
Given two driving scenario descriptions, one is the reference description, the other is the output description from a model. Please extract key information from the reference description and use it as comparison standards to compare the proximity of the output descriptions. Additionally, evaluate the completeness of each piece of information. Key information extracted can be classified into environmental conditions and critical events. Environmental condition information includes weather, time, road environment, and lane conditions. A critical event describes the event related to ego vehicleś driving decisions, which is usually given in the format of critical object + verb/adj + influence on ego vehicle. An example of a critical event is like this: A construction zone is blocking the ego lane and the right lane in front, so the car needs to go around to avoid it. Now I will give you the reference description and the output description.
The reference description:
Weather: Sunny. —— Time: Day. —— Road Environment: Urban. —— Lane Options: Left Lane, Own Lane, Right Lane. —— Ego Lane Position: Middle Lane.
Critical Events: There are police officers ahead inspecting passing vehicles. The police officer in front of your lane is signaling you to stop for inspection. There are also vehicles queuing in the left lane. The right lane is a merging lane and is currently separated, so you cannot change lanes to the right.
The output description:
Weather: Sunny. —— Time: Day.
There is a vehicle driving in the distance in front of you. There are traffic police on the left and right, and the traffic police signal to stop.
Compare the two descriptions using extracted key information and output a similarities score on the scale from 0 to 1.0 and an explanation to describe the logic of calculating the similarity scale. Each matched key information scores 1.0, and each partially matched key information scores 0.5. Hallucinations will be penalized as well. If any key information exists in the output but not in the reference, a score of -0.25 will be assigned. The final score is the total score divided by the number of key information extracted in the reference description. For example, if the total score is 1.5, and the total number of key information in reference description is 3, then the final score would be 1.5/3 = 0.5. Please output the detailed extraction and evaluation process.

**Response:**
Reference description key information:
1. Weather: Sunny (environmental conditions)
2. Time: Day (environmental conditions)
3. Road Environment: Urban (environmental conditions)
4. Lane Options: Left Lane, Own Lane, Right Lane (environmental conditions)
5. Ego Lane Position: Middle Lane (environmental conditions)
6. Police officers ahead inspecting vehicles (critical event)
7. The police officer in front of ego vehicle signalling to stop (critical event)
8. Vehicles queuing in the left lane (critical event)
9. Cannot change to the right lane as itś an merging lane and currently separated (critical event)
Output description key information:
1. Weather: Sunny.
2. Time: Day.
3. Vehicle driving in the distance in front of driver
4. Traffic police on the left and right
5. Traffic police signal to stop
By comparing the output to the reference:
- Key information 1, 2, 6, and 7 are matched, each for 1.0 points(weather and time, police officers and their signal to stop)
- Key information 3 in the output is not mentioned in reference, deduction of -0.25 points.
No scores for:
- Key information 3, 4, 5, 8, and 9.
Total score = (1.0 + 1.0 + 1.0 + 1.0 - 0.25) / 9 = 0.417.

Table 5: **Evaluation process of scene description by GPT-4.** Our proposed evaluation method is not only capable of extracting and evaluating information from structured scene descriptions but is also applicable to unstructured descriptions.

both descriptions. Lastly, the prompt outlines the criteria for scoring, as well as the format for the evaluation output, ensuring a structured and systematic approach to the evaluation process.

## B.2 Meta-action Evaluation

The evaluation process for the meta-action sequence must consider both the quantity and the sequential arrangement of the matched meta-actions. We employ dynamic programming to compare the model's output and the annotated ground truth. Our dynamic programming approach is similar to the method utilized in identifying the longest common subsequence, albeit with two supplementary considerations.

The first consideration acknowledges the unequal weighting of different meta-actions. For instance, certain meta actions such as "*Slow Down*", "*Wait*", and "*Go Straight Slowly*" exhibit a greater emphasis on attitude rather than action. The presence or absence of these actions from a meta-action sequence does not alter the basic semantic essence of driving decisions but rather modifies the driv-

ing strategy to be either more assertive or more cautious. For example, a meta action sequence of "*Slow Down -> Stop -> Wait*" conveys a similar driving decision as a sequence with only the meta action "Stop". Consequently, these sequences should not incur a penalty comparable to other meta actions such as "Turn Left" or "Change Lane to the Right". Therefore, these are designated as "conservative actions", and a reduced penalty is applied when they do not match during sequence evaluation.

The second consideration addresses the potential semantic equality among different meta-action sequences. For example, the sequences "*Change Lane to the Left -> Speed Up -> Go Straight At a Constant Speed -> Change Lane to the Right*" and "*Change Lane to the Left -> Speed Up Rapidly -> Go Straight At a Constant Speed -> Change Lane to the Right*" might both represent valid approaches to overtaking a slow-speed vehicle ahead. Recognizing that different meta-action sequences might convey similar meanings, we initially use GPT-4 to generate variant sequences that have comparable semantic meanings, in addition to the unique ground truth meta-action sequence, as shown in Table 6. In the subsequent sequence-matching phase of the evaluation, all these variations, together with the manually annotated ground truth, are taken into consideration. The highest-scoring matching is then adopted as the definitive score for the final decision evaluation.

---

**Prompt:**
Given a predefined set of action tokens and a sequence of reference driving actions, find all alternative sequences of actions that have similar meanings but choose slightly different action tokens from the predefined action tokens.
Predefined action tokens: [Speed up, Slow down, Speed up rapidly, Slow down rapidly, Go straight slowly, Go straight at a constant speed, Turn left, Turn right, Change lane to the left, Change lane to the right, Shift slightly to the left, Shift slightly to the right, Stop, Wait, Turn around, Reverse].
For example, if the reference driving action sequence is: Slow down -> Shift slightly to the right -> Go straight at a constant speed, alternative action sequences with similar meanings could be:
1. Slow down -> Change lane to the right -> Go straight at a constant speed;
2. Slow down rapidly -> Shift slightly to the right -> Go straight at a constant speed;
3. Slow down -> Change lane to the right -> Go straight slowly;
4. Slow down -> Shift slightly to the right -> Go straight slowly.
Now the reference action sequence is: Slow down -> Shift slightly to the right -> Go straight at a constant speed. What are the alternative action sequences?

---

**Response:**
Given the reference sequence: "Slow down -> Shift slightly to the right -> Go straight at a constant speed." Alternative action sequences with similar meanings based on the provided action tokens are:
1. Slow down -> Change lane to the right -> Go straight at a constant speed.
2. Slow down rapidly -> Shift slightly to the right -> Go straight at a constant speed.
3. Slow down -> Change lane to the right -> Go straight slowly.
4. Slow down -> Shift slightly to the right -> Go straight slowly. You've provided this example in your question.
Based on the tokens and the context, these are indeed appropriate alternative sequences.

---

Table 6: **Example of generating alternative action sequences by GPT-4.**

The state of dynamic programming is saved in a 2D matrix, wherein each row corresponds to a meta action in the ground truth action sequence, and each column corresponds to a meta action in the model output action sequence, noted as $S^{r,c}$. The dynamic programming initiates recursive calculations beginning from the first meta action of both sequences. Each element of the 2D matrix encompasses the optimal total score at the current matching position, as well as the preceding matching condition that yielded the optimal matching. In our dynamic programming algorithm, three transition equations govern distinct cases: $S_{\text{missing}}$ for missing matching, $S_{\text{redundant}}$ for redundant matching, and $S_{\text{matching}}$ for successful matching. Successful matching occurs when the meta action is identical at the $r^{th}$ position in the reference sequence and the $c^{th}$ position in the model-generated sequence. In the case of missing matching, the meta action at the $r^{th}$ position in the reference sequence is unmatched, prompting a comparison with the $r-1^{th}$ position in the reference sequence and the $c^{th}$

position in the model-generated sequence. Conversely, redundant matching implies that the meta action at the $c^{th}$ position in the model-generated sequence is unmatched, leading to further examination of the $r^{th}$ position in the reference and the $c-1^{th}$ position in the model-generated sequence. The transformation equations for these cases are as follows:

$$\begin{aligned}
S_{\text{missing}}^{r,\,c} &= S^{\,r-1,\,c} - p_{\text{missing}}, \\
S_{\text{redundant}}^{r,\,c} &= S^{\,r,\,c-1} - p_{\text{redundant}}, \\
S_{\text{matching}}^{r,\,c} &= S^{\,r-1,\,c-1} + s_{\text{matching}}, \\
S^{\,r,\,c} &= \max(S_{\text{missing}}^{r,\,c}, S_{\text{redundant}}^{r,\,c}, S_{\text{matching}}^{r,\,c}),
\end{aligned} \tag{3}$$

where $s_{\text{matching}} = 1.0$ represents the reward score after a successful matching. If an action considered missing or redundant is classified as a conservative action, the penalties $p_{\text{missing}}$ and $p_{\text{redundant}}$ are quantified as half of $s_{\text{matching}}$, i.e., 0.5. Conversely, if an action is not conservative, both penalties are assigned the same magnitude as $s_{\text{matching}}$, i.e., 1.0. This approach is based on the premise that omitting a crucial meta action or inaccurately introducing a non-existent one equally hampers the effectiveness of the action sequence. The final score $Score_{\text{action}}$ should be divided by the length of the selected reference meta-action sequence, formulated as follow:

$$Score_{\text{action}} = \frac{S^{\,r,\,c}}{N_r} \tag{4}$$

## C  Co-tuning

To preserve the LLM's generalization capabilities during the fine-tuning process, we employed co-tuning with several additional datasets. These include the Talk2Car [59], BDDX [60], Drama [61], SUTD [62], and LLAVA [63] datasets. For each dataset, we conducted random sampling in a 1:1 ratio corresponding to the data volume of the SUP-AD and nuScenes datasets. Following this co-tuning approach, we found that the scores on the SUP-AD dataset, under the evaluation metrics of Scene Description and Meta Action, remained virtually unchanged, simultaneously ensuring the preservation of the LLM's original capabilities and its generalization capacity.

## D  DriveVLM-Dual Onboard Deployment

| Base LLM | Avg. | MMMU 5% | SEEDBench 20% | RefCOCO 15% | SUP-AD 15% | Drivelm-QA 7.5% | Drivelm-Grounding 7.5% | Realworld-VQA 15% |
|---|---|---|---|---|---|---|---|---|
| MobileLLaMA1.4B [55] | 0.457 | 0.331 | 0.59 | 0.421 | 0.520 | 0.686 | 0.735 | 0.501 |
| Qwen-1.8B [64] | 0.477 | 0.340 | 0.622 | 0.492 | 0.523 | 0.680 | 0.725 | 0.518 |
| Gemma-2B [65] | 0.439 | 0.345 | 0.571 | 0.330 | 0.510 | 0.680 | 0.721 | 0.507 |
| MiniCPM-2.4B [66] | 0.482 | 0.379 | 0.64 | 0.444 | 0.539 | 0.676 | 0.717 | 0.553 |
| MobileLLaMA2.7B [55] | 0.496 | 0.348 | 0.635 | 0.557 | 0.546 | 0.683 | 0.725 | 0.536 |
| Phi3-3.8B [67] | 0.538 | 0.435 | 0.688 | 0.608 | 0.604 | 0.697 | 0.743 | 0.592 |
| Qwen-4B [64] | 0.511 | 0.366 | 0.671 | 0.603 | 0.515 | 0.681 | 0.735 | 0.562 |
| Qwen-4B* | 0.529 | 0.373 | 0.684 | 0.624 | 0.596 | 0.699 | 0.738 | 0.553 |

Table 7: Performance of different LLMs on various datasets using the LLAVA-1.5 [68] architecture with ViT-L-336 [69] as the image encoder. Note that * indicates using SigLIP-L-384 [54] as the image encoder.

**Base LLM**  Due to the limited memory and bandwidth of the vehicle's hardware, we cannot use overly large LLMs to maintain real-time inference. Therefore, we chose models with fewer than 4 billion parameters. As shown in Table 7 and 8, our experiments revealed that on the Orin architecture, the "wide and shallow" Qwen series (wider and fewer layers) models outperform "narrow and deep" models (narrower and more layers) in inference speed.

| LLM | Prompt Length(toks) | Prefill latency (s) | Prefill (tok/s) | Decode (tok/s) | Output (toks) | Decode latency (s) | Model Size (GB) | Layer Num | Head Size | Vocab Size |
|---|---|---|---|---|---|---|---|---|---|---|
| Gemma-2B [65] | 1063 | 0.95 | 1121 | 40.9 | 59 | 1.44 | 4.7 | 18 | 256 | 256000 |
| Phi3-4k [67] | 1045 | 1.3 | 797.3 | 49 | 59 | 1.2 | 7.2 | 32 | 96 | 32064 |
| MobileLLaMa-2.7B [55] | 1047 | 0.92 | 1134 | 61.7 | 59 | 0.96 | 5.0 | 32 | 80 | 32000 |
| MobileLLaMa-1.4B [55] | 1047 | 0.23 | 4634 | 117.4 | 59 | 0.5 | 2.5 | 24 | 128 | 32000 |
| Qwen4B [64] | 1078 | 0.57 | 1882 | 44.5 | 59 | 1.33 | 7.5 | 40 | 128 | 151936 |
| Qwen1.8B [64] | 1078 | 0.23 | 4709 | 79.6 | 59 | 0.74 | 3.7 | 24 | 128 | 151936 |

Table 8: Inference performance of different LLMs after quantization and deployment on an OrinX chip. The Qwen series achieved the best performance.

| Backbone | Token Length | Method | Weighted Total | MMMU 5% | SEEDBench 20% | RefCOCO 15% | SUP-AD 15% | Drivelm-QA 7.5% | Drivelm-Grounding 7.5% | Realworld-VQA 15% | PointVQA 15% |
|---|---|---|---|---|---|---|---|---|---|---|---|
| ViT-L-336 [69] | - | - | 0.421 | 0.368 | 0.657 | 0.502 | 0.553 | 0.688 | 0.742 | 0.541 | 0.583 |
| ViT-L-336 | - | Fixed 1x2 + CA-256 | 0.413 | 0.388 | 0.62 | 0.327 | 0.543 | 0.699 | 0.733 | 0.536 | 0.563 |
| ViT-L-336 | - | Fixed 2x2 + CA-256 | 0.405 | 0.376 | 0.618 | 0.306 | 0.518 | 0.692 | 0.731 | 0.523 | 0.594 |
| ViT-L-448 | - | Fixed 1x2 + CA-256 | 0.383 | 0.391 | 0.542 | 0.184 | 0.528 | 0.689 | 0.718 | 0.47 | 0.562 |
| ViT-L-336 | - | Fixed 1x2 + S2 | 0.368 | 0.383 | 0.511 | 0.198 | 0.469 | 0.689 | 0.718 | 0.469 | 0.314 |
| ViT-L-336 | - | Fixed 1x2 + S2 | 0.368 | 0.381 | 0.537 | 0.207 | 0.544 | 0.698 | 0.729 | 0.46 | 0.554 |
| ViT-L-336 | - | DM-4 + CA-256 | 0.409 | 0.377 | 0.61 | 0.277 | 0.544 | 0.698 | 0.726 | 0.525 | 0.554 |
| ViT-L-336 | - | DM-6 + CA-256 | 0.381 | 0.381 | 0.53 | 0.162 | 0.523 | 0.7 | 0.729 | 0.466 | 0.584 |
| ViT-L-336 | - | DM-4 + PT | 0.426 | 0.376 | 0.653 | 0.557 | 0.585 | 0.7 | 0.743 | 0.54 | 0.598 |
| ViT-L-336 | - | DM-4 + SM + LT | 0.413 | 0.403 | 0.617 | 0.293 | 0.55 | 0.699 | 0.737 | 0.527 | 0.532 |
| ViT-L-336 | - | DM-4 + SM + AAP + LT | 0.406 | 0.384 | 0.61 | 0.273 | 0.518 | 0.68 | 0.726 | 0.52 | 0.533 |
| ViT-L-336 | - | DM-4 + SM + CD + LT | 0.409 | 0.388 | 0.63 | 0.48 | 0.515 | 0.7 | 0.734 | 0.524 | 0.577 |
| SigLIP-L-384 [54] | 576 | - | 0.432 | 0.389 | 0.631 | 0.615 | 0.624 | 0.707 | 0.749 | 0.556 | 0.56 |
| SigLIP-L-768 | 576 | - | 0.438 | 0.377 | 0.642 | 0.58 | 0.628 | 0.712 | 0.764 | 0.561 | 0.556 |
| SigLIP-L-1152 | 576 | - | 0.436 | 0.377 | 0.637 | 0.595 | 0.628 | 0.715 | 0.763 | 0.571 | 0.564 |
| SigLIP-L-384-768 | 576 | PE | 0.434 | 0.367 | 0.632 | 0.581 | 0.631 | 0.716 | 0.762 | 0.556 | 0.554 |
| SigLIP-L-512-960 | 480 | PE | 0.442 | 0.369 | 0.64 | 0.557 | 0.65 | 0.719 | 0.762 | 0.579 | 0.568 |

Table 9: Performance of different methods for scaling ViT's original input resolution to higher resolution. "CA" stands for cross-attention, "DM" is the abbreviation for Dynamic Max, "PT" indicates the use of patch end token, "SM" stands for Spatial Merge, "LT" indicates the use of line end token, "AAP" is the abbreviation for Adaptive Average Pooling, and "CD" stands for Convolutional Downsampling. "PE" represents Position Embedding Interpolation. Among these methods, applying PE interpolation to SigLIP-L-384, transforming it to accept 768 resolution images as input, achieves a good trade-off between inference speed and performance.

**Visual Encoder**   High-resolution images are essential for fine-grained understanding in autonomous driving. As shown in Table 9, compared to the basic ViT model used as a visual encoder, we explored several options, including different GridPatch strategies and PE (Position Embedding) interpolation. Ultimately, for real-time inference, we selected the simpler SigLIP-L-384 model with PE interpolation, achieving high resolution input through original 384 resolution PE interpolation and fine-tuning parameters with additional conv layers.

**Visual Token Compression**   To address the increased computational load from high-resolution images, we implemented LDPNetv2 [70] to reduce the number of image tokens by 75% without compromising performance, as shown in Table 10. Additionally, we enhanced performance by replacing avg-pooling layer with convolutional layer in LDPNetv2.

**Speculative Sampling**   Speculative Sampling is used to accelerate inference by preemptively generating likely outputs. This approach reduces the latency of generating predictions, achieving a significant speedup without substantial loss in accuracy. As shown in Table 11, we test two speculative sampling method Medusa and Eagle with our inference framework designed specifically for OrinX chip. Eagle achieved a 2.7x speedup in decode latency compared to Medusa's 2.17x, making real-time vehicle deployment feasible.

| Projector | Origin | Compressed | Output(ms) | Prefill(ms) | Avg. | MMMU | SEEDV2 | BDD | Drivelm-QA | Drivelm-grounding | Realworld-VQA | RefCOCO | SUP-AD |
|---|---|---|---|---|---|---|---|---|---|---|---|---|---|
| MLP (Baseline) | 576 | 576 | 666.60 | 707.93 | 56.27 | 37.90 | 64.00 | 53.90 | 67.60 | 71.70 | 55.30 | 44.40 | 53.90 |
| LDPNetV2 [70] | 576 | 576 | 661.70 | 707.42 | 50.53 | 37.67 | 56.71 | 53.36 | 68.90 | 73.20 | 48.30 | 21.29 | 54.70 |
| Perceiver Resampler [71] | 576 | 576 | 637.20 | 666.20 | 49.71 | 39.93 | 58.84 | 52.21 | 68.37 | 71.90 | 49.61 | 19.01 | 48.70 |
| Pixel shuffle | 576 | 256 | 610.70 | 655.92 | 52.02 | 40.42 | 60.20 | 55.19 | 68.72 | 71.52 | 51.44 | 28.23 | 48.40 |
| LDPNetV2 | 576 | 256 | 605.32 | 652.79 | 54.73 | 38.93 | 62.19 | 54.82 | 68.57 | 72.78 | 50.59 | 34.32 | 59.30 |
| Pixel shuffle | 576 | 144 | 604.98 | 645.61 | 49.40 | 38.98 | 61.63 | 56.06 | 69.18 | 73.18 | 51.05 | 30.67 | 58.80 |
| LDPNetV2 | 576 | 144 | 597.15 | 646.77 | 55.56 | 38.93 | 62.63 | 56.23 | 68.93 | 72.48 | 50.59 | 33.14 | 59.30 |
| LDPNetV2 | 576 | 64 | 616.20 | 645.18 | 56.24 | 39.40 | 61.80 | 54.60 | 68.80 | 72.60 | 51.00 | 41.88 | 61.20 |

Table 10: Performance of different methods for visual token compression. Using LDPNetV2 to compress the original tokens to 75% of the original token count, achieves the best trade-off between performance and speed.

| | Base + q4f16_1 | Eagle [58] + q4f16_1 | Medusa [72] + q4f16_ft | Eagle + q4f16_ft | Eagle + q4f16_ft + Shrink Vocab Size(1024) |
|---|---|---|---|---|---|
| Quant Type | q4f16_1 | q4f16_1 | q4f16_ft | q4f16_ft | q4f16_ft |
| Input Size | (384, 960) | (384, 960) | (384, 960) | (384, 960) | (384, 960) |
| Prefill Tokens | 604 | 604 | 613 | 604 | 613 |
| Output Tokens | 37 | 39 | 41 | 41 | 41 |
| Prefill Speed (tok/s) | 1818 | 1793 | 2160.64 | 2230 | 2175.35 |
| Decode Speed (tok/s) | 109 | 172 | 232.34 | 295.04 | 518.3 |
| Prefill Latency (s) | 0.332 | 0.340 | 0.284 | 0.276 | 0.274 |
| Decode Latency (s) | 0.328 | 0.216 | 0.176 | 0.130 | 0.071 |
| Acceleration ratio | 1 | 1.57 | 2.17 | 2.7 | 4.33 |

Table 11: Performance of different speculative sampling method. "Shrink Vocab Size" means we reduce the vocabulary to the 1024 most frequently used words. "q4f16_1" is a 4-bit quantization method using a 16-bit floating-point representation for efficient model compression, while "q4f16_ft" includes subsequent fine-tuning to enhance performance post-quantization.

# E   Qualitative Results

To further demonstrate the effectiveness and robustness of our DriveVLM, we provide additional visualization results in Figure 14, 15, 16, 17, and 18. In Figure 14, DriveVLM recognizes the slowly moving vehicle ahead and provides a driving decision to change lanes for overtaking. In Figures 15 and 16, DriveVLM accurately identifies the type of unconventional vehicles and a fallen tree, demonstrating its capability in recognizing long-tail objects. In Figure 17, the traffic police signaling to proceed with hand gestures has been accurately captured by DriveVLM. In Figure 18, DriveVLM successfully recognizes the road environment of a roundabout and generates a planned trajectory with a curved path.

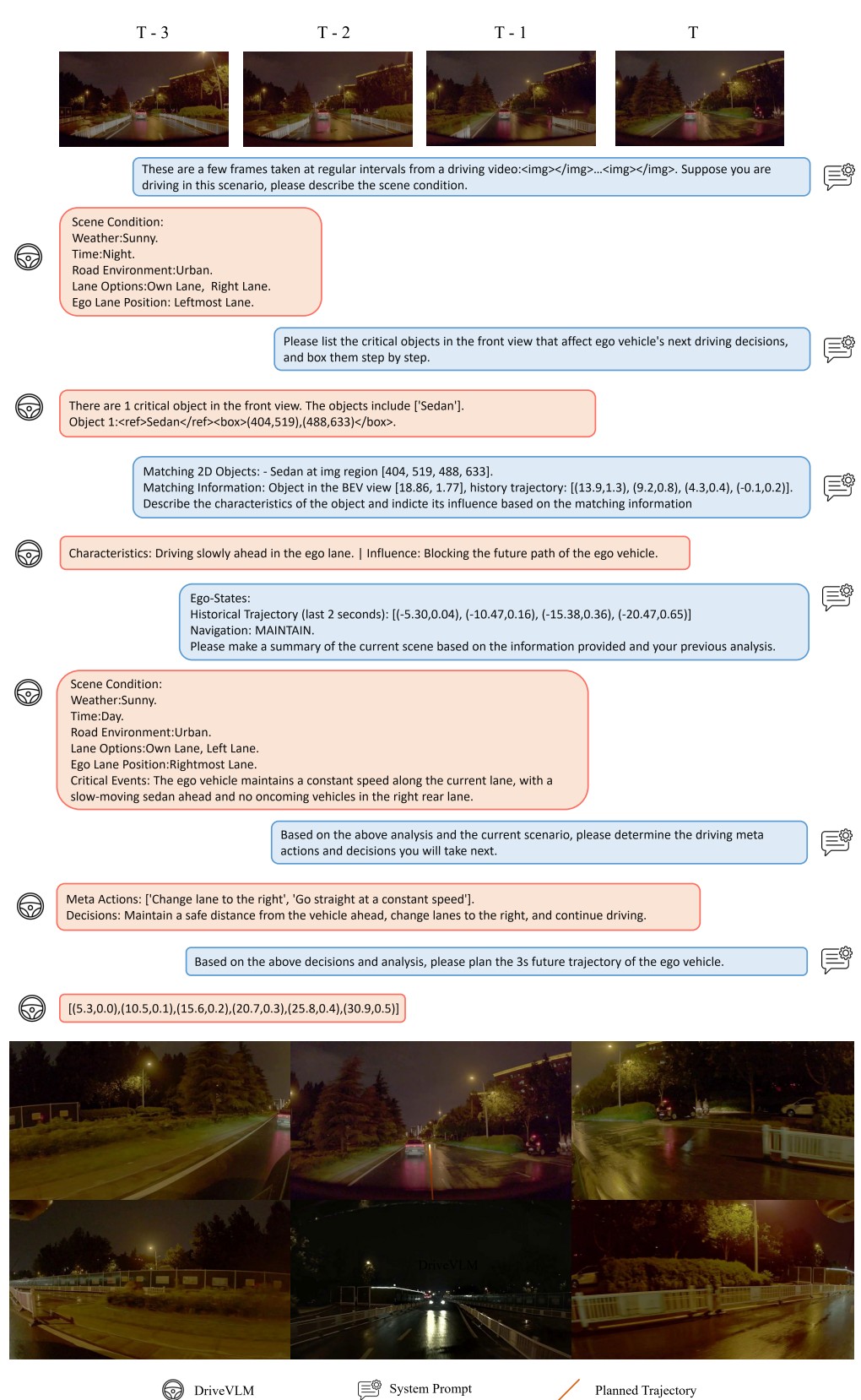

Figure 14: **Visualization of DriveVLM's output.** DriveVLM recognizes the slowly moving vehicle ahead and provides a driving decision for changing lanes to overtake.

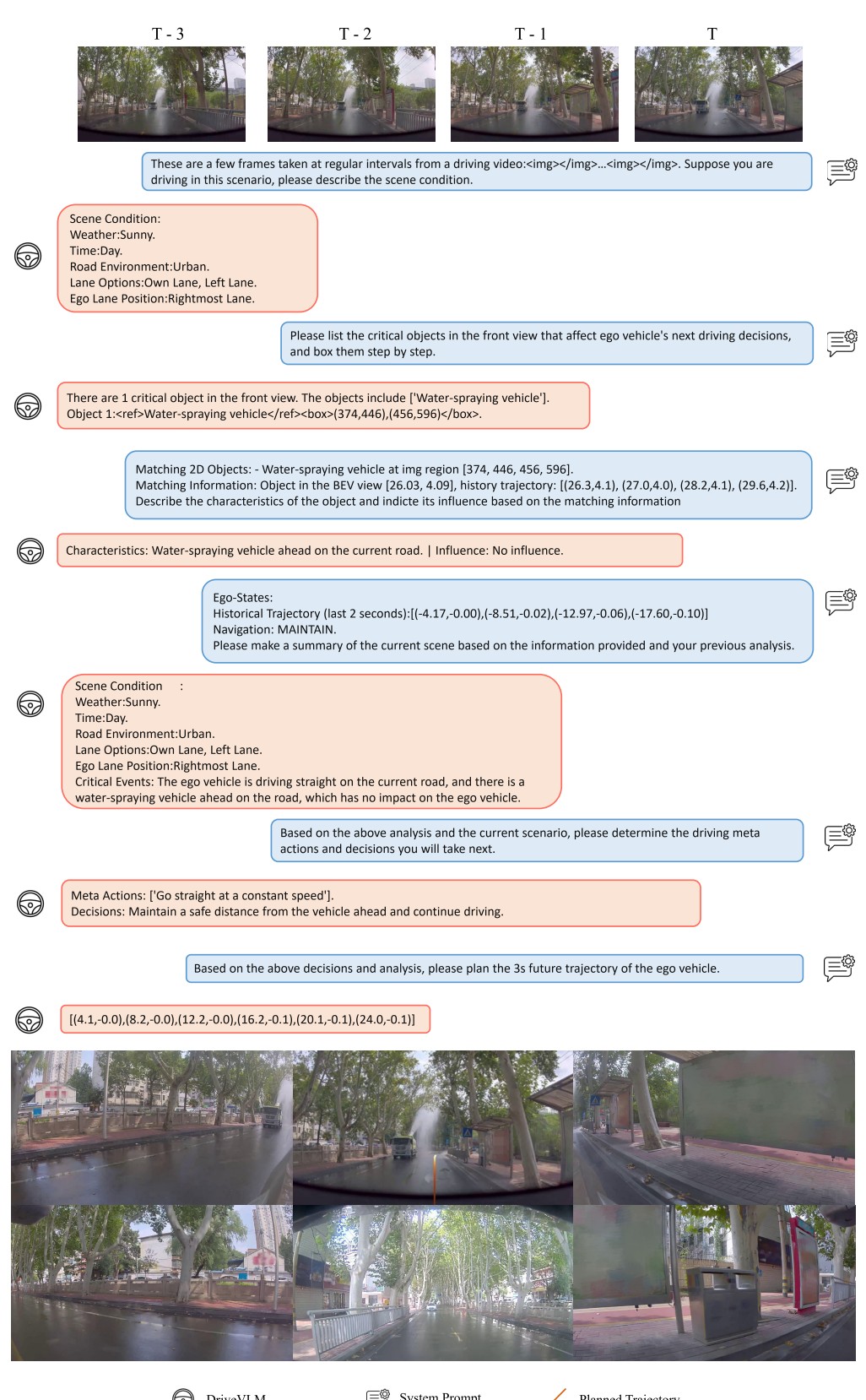

Figure 15: **Visualization of DriveVLM's output.** DriveVLM has identified the type of the unconventional vehicle ahead.

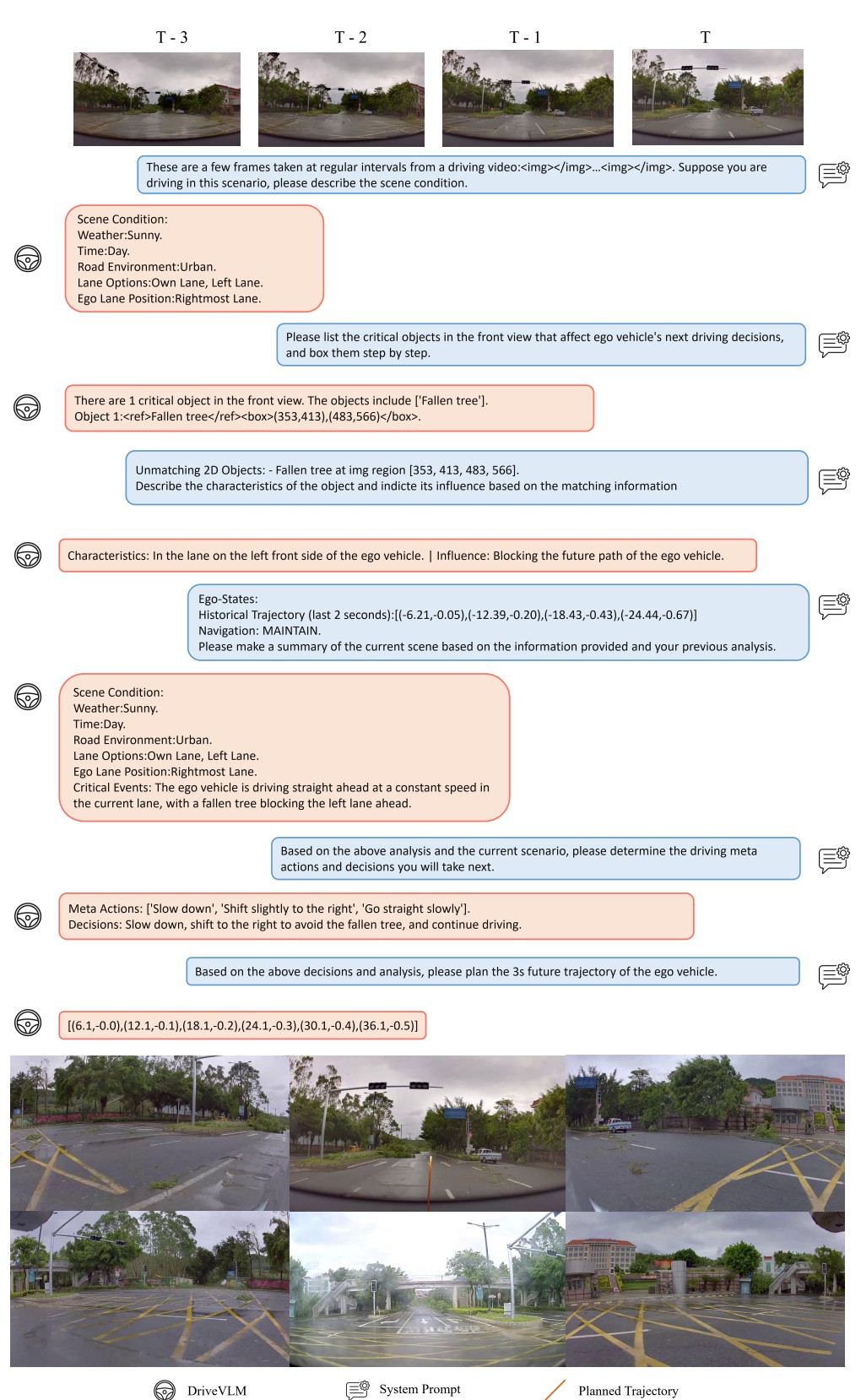

Figure 16: **Visualization of DriveVLM's output.** DriveVLM precisely detect the fallen tree and its position, subsequently planning an appropriate detour trajectory.

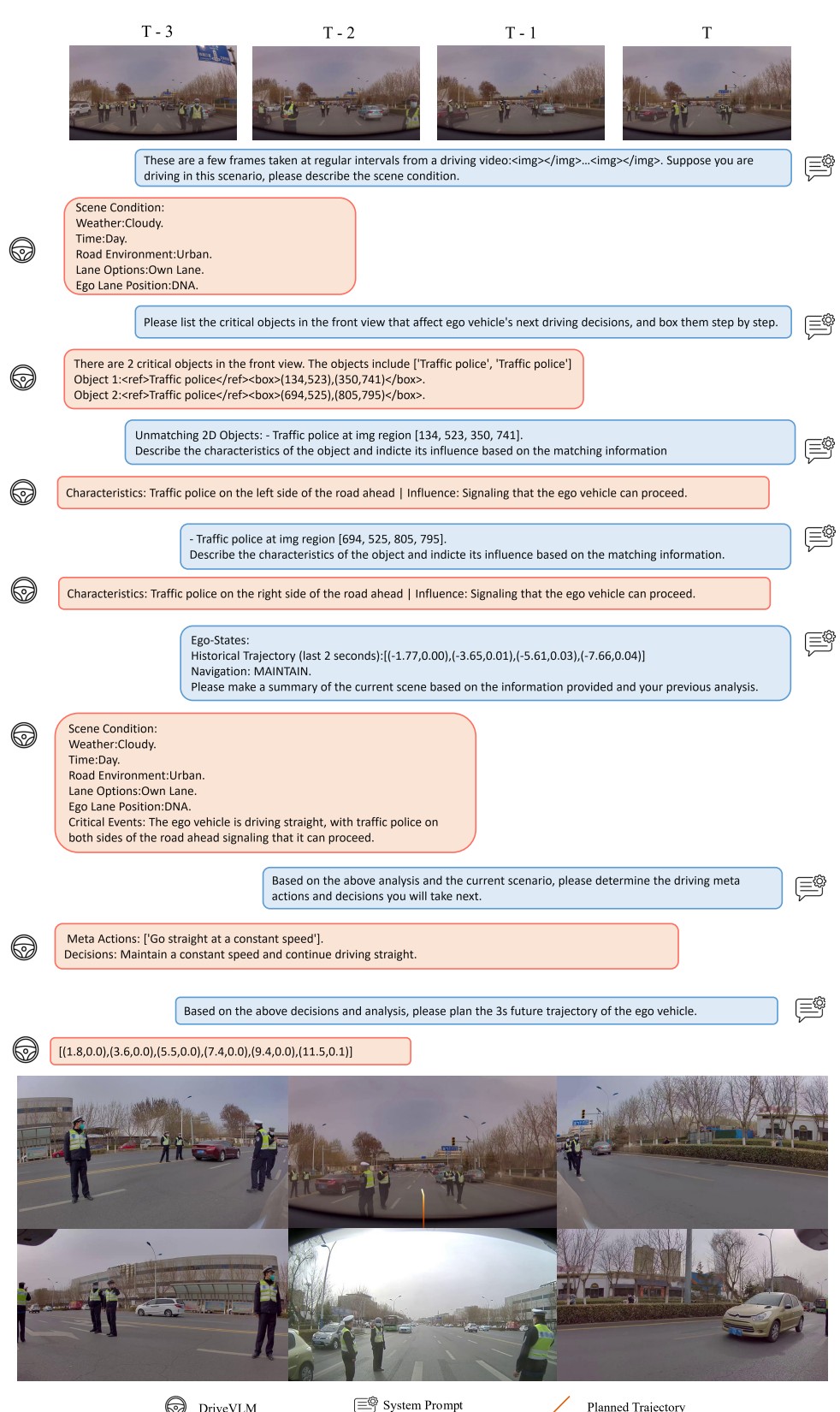

Figure 17: **Visualization of DriveVLM's output.** The traffic police signaling to proceed with hand gestures has been accurately captured by DriveVLM.

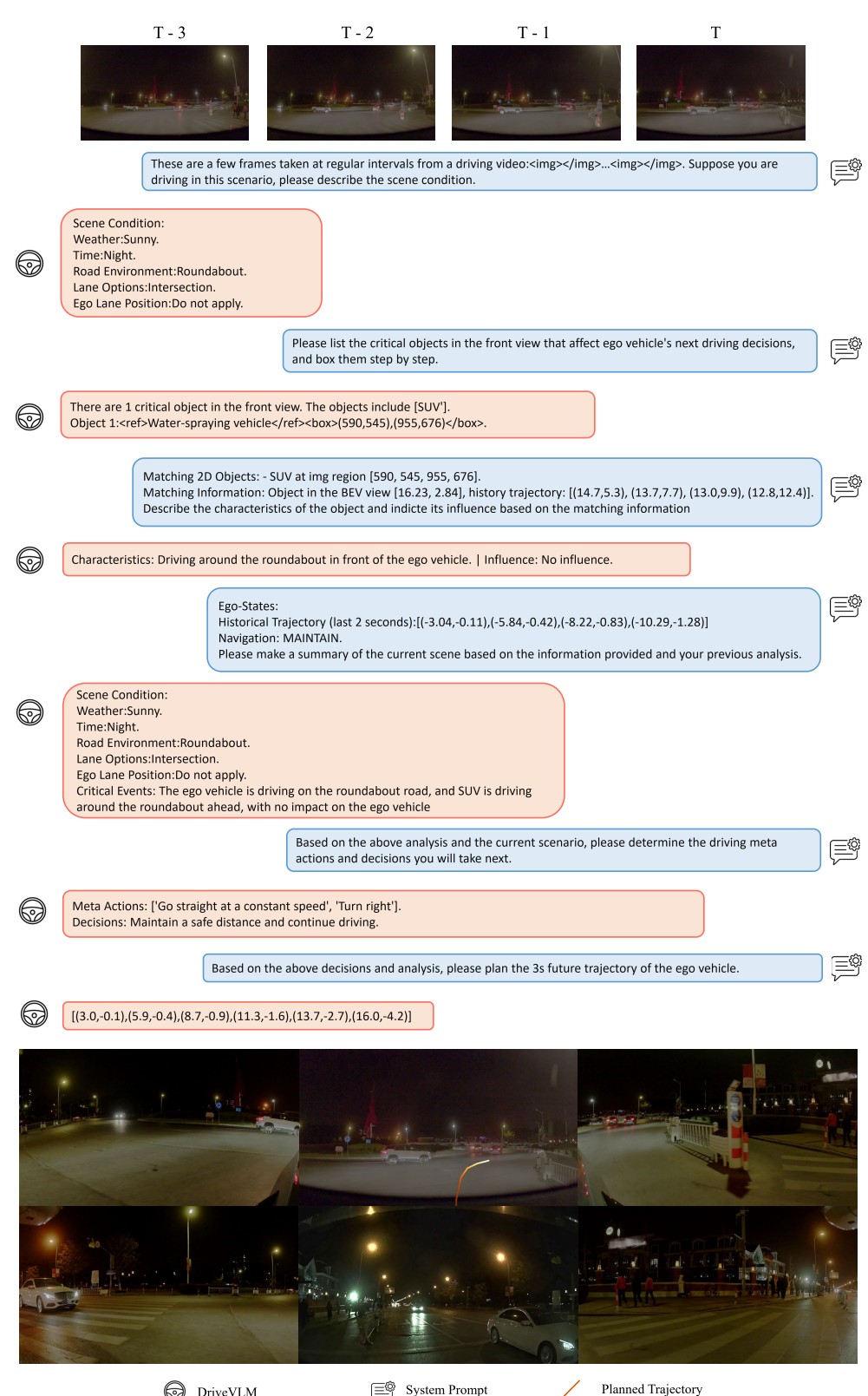

Figure 18: **Visualization of DriveVLM's output.** DriveVLM successfully recognizes the road environment of a roundabout and generates a planned trajectory with a curved path.