# OpenReview forum: "DriveVLM: The Convergence of Autonomous Driving and Large Vision-Language Models"
_robot-learning.org/CoRL/2024/Conference — CoRL 2024_

### Official Review · Reviewer_1XCj · 2024-07-17
**Paper review of submission 592**

**Originality:** 4
**Technical Quality:** 3
**Clarity Of Presentation:** 3
**Potential Impact:** 3
**Recommendation:** 3
**Confidence:** 4

**Review:**

Strengths:
* Introduces DriveVLM and DriveVLM-Dual, which leverage Vision - Language Models (VLMs) to enhance scene understanding and planning in autonomous driving, providing a novel solution to address the long tail challenges in this field.
* Proposes the DriveVLM - Dual hybrid system that combines the strengths of DriveVLM and the traditional autonomous pipeline, improving spatial reasoning and real-time planning capabilities.
* Defines a scene understanding for planning task, along with evaluation metrics, and constructs a dataset (SUP-AD) through a data mining and annotation pipeline, which is valuable for research in this area.
* Demonstrates the superior performance of DriveVLM and DriveVLM-Dual in experiments on both the nuScenes dataset and the SUP-AD dataset, and verifies their effectiveness through onboard deployment and testing.

Weaknesses:
* It would be more desirable if System 1 could acquire additional information from the VLM. For an optimization-based planner, only a reference trajectory is sufficient; however, for a neural-based planner, more information could be shared through the feature space.
* While the paper acknowledges the limitations of VLMs in spatial grounding and reasoning, and the high latency and computational intensity, a more detailed discussion on the potential limitations and challenges of the proposed system could be beneficial.

**Quality Of The Limitations Section:**

1

**Questions For Rebuttal:**

* How to handle the situation where one of the systems fails or makes significantly different decisions?
* In the environment description, whether the extreme weather conditions will be taken into consideration, which might be beneficial for the end-to-end driving system.
* In Appendix part, you mentioned meta actions are classified into 17 categories, but why there is only 15 categories in Figure 1 ?

**Robotics Focus:**

4

**Summary Of Paper:**

This paper introduces DriveVLM, an autonomous driving system that leverages Vision-Language Models (VLMs) for enhanced scene understanding and planning capabilities. The authors also propose DriveVLM-Dual, a hybrid system that combines the strengths of DriveVLM and the traditional autonomous driving pipeline.

**Summary Of Recommendation:**

The paper is well organized with sufficient hardware demonstration. So I recommend a weak accept for this paper.

---

### Official Review · Reviewer_fMiT · 2024-07-22
**Paper Review**

**Originality:** 3
**Technical Quality:** 3
**Clarity Of Presentation:** 3
**Potential Impact:** 3
**Recommendation:** 3
**Confidence:** 4

**Review:**

Strengths:
1. The paper is generally well-written and easy to follow.
2. The paper provides thorough experimental results on both the nuScenes dataset and a custom SUP-AD dataset, showcasing the systems' capabilities.
3. The demo video provides interesting real-world demonstrations of the proposed method that were deployed on a real vehicle.

Weaknesses:
1. The proposed method involves a large VLM, which requires a huge amount of computational resources, which may lead to latency in practical deployment. However, there seems no explicit discussion on the running time of the model and the effect of inference latency on the end-to-end framework performance. How much does the frequency of the "low-frequency" component influence the performance? It would be great to provide more discussions on the computational efficiency and the required computational resources onboard autonomous vehicle of the proposed framework.
2. The authors claim that DriveVLM can identify long-tail critical objects. It would be better to provide a more concrete definition of the long-tail critical objects. How you decide which objects are long-tail ones? It would be better to provide quantitative and qualitative results on the evaluation of long-tail cases?
3. There is no discussion on the limitations of the proposed framework.

**Quality Of The Limitations Section:**

1

**Questions For Rebuttal:**

Please address the comments and questions in the "Review" section during the rebuttal phase. In particular, it would be great to provide more experimental results mentioned in the comments.

**Robotics Focus:**

4

**Summary Of Paper:**

The authors introduce DriveVLM, an autonomous driving system leveraging VLMs for enhanced scene understanding and planning capabilities. The proposed method integrates a unique combination of reasoning modules for scene description, scene analysis, and hierarchical planning. Furthermore, recognizing the limitations of VLMs in spatial reasoning and heavy computational requirements, the authors propose DriveVLM-Dual, a hybrid system that synergizes the strengths of DriveVLM with the traditional autonomous driving pipeline.

**Summary Of Recommendation:**

More experimental results and discussions are required to demonstrate the effectiveness of the proposed framework, which are elaborated in the "Review" section.

---

### Official Review · Reviewer_cCjc · 2024-07-23
**Additional planning evaluation on long-tail scenarios could further improve the paper.**

**Originality:** 3
**Technical Quality:** 3
**Clarity Of Presentation:** 3
**Potential Impact:** 3
**Recommendation:** 3
**Confidence:** 3

**Review:**

Strengths:

1. This paper is well written, and the logic is clear.

2. The annotated dataset could greatly help the VLM for AV community if the authors plan to release the Nuscenes dataset.

3. The proposed DriveVLM dual is validated using onboard experiment

Weakness:

Q1. It has been shown that using ego state along can achieve SOTA performance in the Nuscenes dataset. I am wondering how the driveVLM performs without the ego state information.

Q2. I think the main reason for using LLM for driving is that it contains reasonably good knowledge about how to drive in complex long-tail settings that traditional ML-based planner may struggle with. Even though the authors spent effort on annotating long-tail scenarios,  the planning evaluation seems to only focuses on the entire validation dataset. This makes it difficult to evaluate can LLM truly help planning in long-tail settings. Since the authors already annotated and identified these critical driving scenarios, it would be great if the L2 and collision metrics on these long-tail settings could be reported.

**Quality Of The Limitations Section:**

1

**Questions For Rebuttal:**

See Q1 and Q2 in the main review.

**Robotics Focus:**

3

**Summary Of Paper:**

This paper studies the use of Vision-Language Models (VLM) to enhance Autonomous Vehicles’ (AV) scene understanding, reasoning, and planning performance. The proposed VLM uses Vision Transformers (ViT) as the image tokenizer and Qwen as the LLM backbone. The Visual Question Answering (VQA) datasets are custom-built using a combination of rules and human labor. The paper proposes two frameworks: DriveVLM - utilizes the proposed VLM for various AV tasks; DriveVLM Dual - Injecting perception information (critical object analysis) from DriveLLM into the traditional AV 3D perception module and using the low-frequency motion plan from DriveVLM as the initial plan in the traditional AV planning module. The DriveVLM Dual framework is validated using onboard experiments.

**Summary Of Recommendation:**

The major contribution is the annotated dataset (if the authors plan to release the annotated Nuscenes dataset) and the validation of driveVLM-dual in real AV systems. Additional planning evaluation results in long-tail scenarios could further improve the paper.

---

### Author Rebuttal · Authors · 2024-08-13

Dear area chair and reviewers

The attachment contain following two items:
1. **Additional visualization results for rebuttal.**
2. **Revised paper**

---

### Decision · Program_Chairs · 2024-09-04

**Decision:**

Accept

**Comment:**

In brief, the strengths of the paper are, the clarity of presentation, good experimental results, and technically sound framework. The Reviewers appreciate the authors effort on adequately addressing the reviewers' comments during the rebuttal stage. The paper has significantly improved its quality.